# Clinical Benefits of Indocyanine Green Fluorescence in Robot-Assisted Partial Nephrectomy

**DOI:** 10.3390/cancers14123032

**Published:** 2022-06-20

**Authors:** Yu-Kuan Yang, Ming-Li Hsieh, Sy-Yuan Chen, Chung-Yi Liu, Po-Hung Lin, Hung-Cheng Kan, See-Tong Pang, Kai-Jie Yu

**Affiliations:** 1Division of Urology, Department of Surgery, Linkou Chang Gung Memorial Hospital, Taoyuan 333, Taiwan; luke820737@cgmh.org.tw (Y.-K.Y.); h0810@cgmh.org.tw (M.-L.H.); mr1711@cgmh.org.tw (S.-Y.C.); m7587@cgmh.org.tw (P.-H.L.); m0320@cgmh.org.tw (H.-C.K.); 2College of Medicine, Graduate Institute of Clinical Medical Sciences, Chang Gung University, Taoyuan 333, Taiwan; 8902087@cgmh.org.tw; 3Department of Urology, New Taipei Municipal Tucheng Chang Gung Memorial Hospital, New Taipei City 236, Taiwan; 4Department of Chemical Engineering and Biotechnology, Graduate Institute of Biochemical and Biomedical Engineering, National Taipei University of Technology, Taipei 106, Taiwan

**Keywords:** robot-assisted partial nephrectomy, renal cell carcinoma, indocyanine green, nephron sparing

## Abstract

**Simple Summary:**

Indocyanine green (ICG) administration in robot-assisted partial nephrectomy (RAPN) can minimize warm ischemia time and preserve more parenchyma resulting in exceptional preservation of renal function and reduced incidence of postoperative complications. However, previous studies have seldom compared how ICG-RAPN use differs when used to treat benign versus malignant renal tumors because the baseline patient and tumor characteristics as well as treatment goals are completely different. The aim of our retrospective study was to compare the intraoperative and postoperative outcomes and the differences in the results of ICG administration between patients with benign and malignant tumors. We have demonstrated that ICG-RAPN yielded superior preservation of short-term renal function. Of the patients with malignant renal tumors, it had less operative blood loss without a more positive margin rate than standard RAPN.

**Abstract:**

Background: To compare the intraoperative and postoperative outcomes of indocyanine green (ICG) administration in robot-assisted partial nephrectomy (RAPN) and report the differences in the results between patients with benign and malignant renal tumors. Methods: From 2017 to 2020, 132 patients underwent RAPN at our institution, including 21 patients with ICG administration. Clinical data obtained from our institution’s RAPN database were retrospectively reviewed. Intraoperative, postoperative, pathological, and functional outcomes of RAPN were assessed. Results: The pathological results indicated that among the 127 patients, 38 and 89 had received diagnoses of benign and malignant tumors, respectively. A longer operative time (311 vs. 271 min; *p* = 0.006) but superior preservation of estimated glomerular filtration rate (eGFR) at 3-month follow-up (90% vs. 85%; *p* = 0.031) were observed in the ICG-RAPN group. Less estimated blood loss, shorter warm ischemia time, and superior preservation of eGFR at postoperative day 1 and 6-month follow-up were also noted, despite no significant differences. Among the patients with malignant tumors, less estimated blood loss (30 vs. 100 mL; *p* < 0.001) was reported in the ICG-RAPN subgroup. Conclusions: Patients with ICG-RAPN exhibited superior short-term renal function outcomes compared with the standard RAPN group. Of the patients with malignant tumors, ICG-RAPN was associated with less blood loss than standard RAPN without a more positive margin rate. Further studies with larger cohorts and prospective designs are necessary to verify the intraoperative and functional advantages of the green dye.

## 1. Introduction

Partial nephrectomy (PN) is the standard surgical treatment for small renal masses, as it has exhibited superior renal functional preservation and long-term patient survival relative to radical nephrectomy [1,2,3,4]. Compared with other methods for PN, robot-assisted PN (RAPN), which has become increasingly prominent in recent years, yields fewer intraoperative and postoperative complications, superior renal functional outcomes, and lower rates of positive margins and conversion to radical nephrectomy than open and laparoscopic PN [5,6,7]. Near-infrared fluorescence (NIRF) using indocyanine green (ICG) has been adopted as a safe and practical tool to identify anatomical structures in both oncological and non-oncological surgeries, helping verify devascularization and indicate resection margins, offering a benefit in terms of preserving short-term renal function [8,9,10,11,12,13,14,15,16,17,18,19,20,21]. Studies have mainly discussed the outcomes of ICG in aiding selective clamping without ischemia time or with reduced warm ischemia time (WIT) [14,17,18,19,22,23]; however, they have seldom compared how ICG-RAPN use differs when used to treat benign versus malignant renal tumors because the baseline patient and tumor characteristics as well as treatment goals are completely different [3,4,24,25]. Therefore, in this study, we not only compared the outcomes of ICG-RAPN and standard RAPN but also reported the differences in the results between patients with benign and malignant tumors.

## 2. Materials and Methods

### 2.1. Patients

From January 2017 through December 2020, 132 patients received RAPN for kidney tumor at a tertiary center and 5 patients were excluded due to off-clamp method; the group included 21 patients who underwent intraoperative administration of ICG. All the patients had completed either a preoperative computed tomography (CT) scan or magnetic resonance imaging to allow health care personnel to examine the anatomical structures of their kidneys and the characteristics of their kidney tumors. We retrospectively reviewed the demographic data, tumor complexity (using RENAL nephrometry scoring), preoperative hemoglobin (Hb), renal function (using serum estimated glomerular filtration rate (eGFR) determined using the CDK-EPI equation), operative variables (WIT, operative time, and estimated blood loss), postoperative outcomes (pathological results, length of admission in days, intraoperative and postoperative complications according to Clavien–Dindo classification (CD)), 1-year recurrence, trifecta achievement (defined as WIT ≤ 25 min, negative surgical margins, and the absence of ≥3a CD complications), Hb at postoperative day 1, and renal function at postoperative day 1 as well as 3- and 6-month follow-ups.

### 2.2. ICG Injection

RAPN was performed using either a fluorescence-capable da Vinci Si system or the Xi surgical system (Intuitive Surgical, Sunnyvale, CA, USA), and the surgeon determined whether to use ICG depending on the appearance of an unfavorable kidney anatomical structure or high RENAL score. After the renal pedicle was controlled with Bulldogs, 3 to 5 mL of ICG (25 mg, dissolved in 10 mL of distilled water, with a final concentration of 2.5 mg/mL) was applied intravenously to confirm renal ischemia with NIRF imaging. The fluorescence could be seen in the main renal vessels after 75 s. Thereafter, unclamped the renal artery for margin identification then clamped the renal artery again before resection.

### 2.3. Statistical Analysis

Statistical analysis was performed with SPSS for Mac, version 25.0 (Chicago, IL, USA). Continuous variables are presented using median values with interquartile ranges (IQR), and categorical variables are presented as counts and proportions (percentages). The Mann–Whitney *U* test and Fisher’s exact test or chi-square analysis were employed as appropriate to compare the intraoperative and postoperative data, with statistical significance defined at *p* < 0.05.

## 3. Results

### 3.1. Patients Characteristics

Among the 132 patients who had undergone RAPN, 5 were excluded due to the use of off-clamp surgery. A total of 127 patients were included in the final analysis; 38 (30%) and 89 (70%) patients had received diagnoses of benign and malignant renal tumors, respectively. Significant differences between the benign and malignant tumor groups in terms of the preoperative and postoperative patient characteristics at baseline were observed. More male patients (*p* < 0.001), more hypertension cases (*p* = 0.003), higher preoperative and postoperative Hb levels, lower preoperative and postoperative eGFR, and smaller tumor size (*p* = 0.001) were recorded in the malignant renal tumors group. Significantly longer operative time (311 vs. 271 min; *p* = 0.006) and superior eGFR preservation rate at 3-month follow-up (90% vs. 85%; *p* = 0.031) were observed in the ICG-RAPN groups. Less estimated blood loss (50 vs. 100 mL; *p* = 0.09), shorter WIT (21 vs. 24 min; *p* = 0.25), and higher postoperative eGFR preservation rate at day 1 and 6-month follow-up were noted in the ICG-RAPN groups, despite no significant differences. Furthermore, no other significant differences were noted in terms of the postoperative outcomes, and no ICG-related adverse effects were recorded. Demographic and preoperative characteristics of our study participants are listed in Table 1, and intraoperative and postoperative variables are summarized in Table 2.

### 3.2. Patients with Benign Renal Tumors

Among the 38 patients with benign renal tumors, 10 received ICG-RAPN, and 28 received standard RAPN. No significant differences in terms of baseline patient characteristics, tumor complexity, or laboratory data were noted between these two groups. However, larger tumor sizes in preoperative images (7.3 cm vs. 4.4 cm; *p* = 0.172) were observed in the ICG-RAPN group despite the difference being nonsignificant. ICG-RAPN entailed a longer operation time (325 vs. 228 min; *p* < 0.001). Greater estimated blood loss was noted in ICG-RAPN, but the difference was nonsignificant. The WIT was 20 min for both groups. Two (20% [of subgroup]) and four (14%) patients reported postoperative complications in the ICG-RAPN and standard RAPN groups, respectively, but none of these complications had a CD classification of ≥3. The ICG-RAPN group exhibited lower day 1 postoperative Hb preservation (87% vs. 89%; *p* = 0.351) but superior postoperative eGFR preservation at day 1 as well as 3- and 6-month follow-ups; nevertheless, these differences were all nonsignificant. Angiomyolipoma (AML) was the most common benign kidney tumor in both the ICG and standard groups, accounting for 90% and 96% of such tumors, respectively. Intraoperative and postoperative variables of benign tumors group are summarized in Table 3.

### 3.3. Patients with Malignant Renal Tumors

Among the 89 patients with malignant renal tumors, 11 received ICG-RAPN, and 78 received standard RAPN. The ICG-RAPN group was 11 years younger, on average, than the standard group, but the age difference was nonsignificant (*p* = 0.059). No significant differences were evident in body mass index (BMI), underlying diseases, preoperative tumor size (in images), RENAL score, or preoperative Hb and eGFR. The ICG-RAPN group exhibited longer operation times (298 vs. 280 min; *p* = 0.25) and shorter WIT (24 vs. 26; *p* = 0.207), but neither difference was significant. Furthermore, estimated blood loss was significantly less in the ICG-RAPN group (30 vs. 100 mL; *p* < 0.001) than in the standard one. Two patients (18%) reported postoperative complications in the ICG-RAPN group, one of which a CD classification of ≥3 (postoperative renal pseudoaneurysm was discovered, and the patient underwent transarterial embolization). Twenty-one patients (27%) reported postoperative complications in the standard RAPN group, five (6%) of whom had complications with a CD classification of ≥3 (one patient underwent reopen surgery after the initial RAPN due to hemogenic shock). The patients with high-grade complications and subsequent interventions are listed in Table 4. The preservation rate of postoperative Hb and eGFR at day 1, 3 months, and 6 months were higher in the ICG-RAPN group but with a nonsignificant difference. No statistically significant differences in terms of length of hospital stay, positive surgical margin rate, or trifecta achievement were observed between these two groups. For the histopathologic type, both the ICG and standard RAPN groups had renal cell carcinoma (RCC), and the clear cell subtype was the most prevalent, appearing in 46% and 73% of patients, respectively. The pathologic stage distribution was pT1a (91%), pT1b (9%) in ICG-RAPN and pT1a (77%), pT1b (17%), pT2a (1%) and pT3a (5%) in standard-RAPN. Intraoperative and postoperative variables of the malignant tumors group are summarized in Table 3.

## 4. Discussion

The main goal of PN is to achieve favorable functional outcomes with oncological safety equivalent to radical nephrectomy when treating small renal tumors [1,2,3,4,24].

The previous study revealed that preoperative renal function, the volume of preserved parenchyma and WIT effect the postoperative short-term renal function outcomes after PN [19]. With the help of ICG-RAPN, it can minimize WIT and preserve more parenchyma resulting in exceptional preservation of renal function and reduced incidence of postoperative complications [12,13,14,15,17,18,19,20,21,22,23,26] Our study also compared the differences in the results of ICG administration between patients with benign and malignant tumors. We have demonstrated that ICG-RAPN yielded superior preservation of short-term renal function. Of the patients with malignant renal tumors, it had less operative blood loss without a more positive margin rate than standard RAPN.

The preoperative characteristics of benign and malignant tumors differ; for instance, RCC, which is the most common malignant tumor, is more common in men and is associated with obesity, hypertension, and chronic kidney disease; our previous study results affirmed these associations [27,28]. Furthermore, the indicators and treatment goals of benign and malignant renal tumors are entirely different. Malignant renal tumors such as RCC are mostly found by accident in cross-sectional images prior to symptom manifestation; once clinical T1 cancer is suspected, the latest guidelines recommend surgical intervention (especially PN) as the first option [3,4]. In addition to preserving renal function, negative surgical margins should always be a priority, and enucleation should even be considered in some situations [3,4].

However, for benign renal tumor, active surveillance represents a reasonable strategy, and surgical intervention is usually performed when malignancy is suspected, increasing the risk of hemorrhage in patients with symptoms or large tumors [29,30,31]. PN remains the first option of surgical intervention, but negative surgical margins are not emphasized because of the low recurrence rate after surgery [29,31]. Therefore, when evaluating the benefit of ICG-RAPN on surgical margins, benign renal tumors should be excluded from the analysis, with many of them having unrecognizable surgical margins, as in the case of 8 of the 38 (21%) such patients enrolled in our study.

ICG-RAPN can assist surgeons with the following three tasks. First, it can be employed for identifying the arterial blood supply of renal tumors and can even be used with selective clamping to minimize the ischemia of normal renal parenchyma [8,9,16]. This is because ICG binds with plasma proteins rapidly after its intravenous application, rendering the perfusion of renal arteries easily visible [16]. Treatment of larger tumors, especially malignant ones, can benefit from this method substantially because such tumors tend to have more accessory and complex blood supplies that are difficult to thoroughly identify with preoperative imaging [15]. In our study’s malignant tumor group, the estimated blood loss in the ICG-RAPN subgroup (30 mL) was significantly less than that of the benign tumor group and was much less than reported in previous studies (range: 75–300 mL) [14,15,16,18,19,22,23]. For the high-grade postoperative complications like re-open surgery due to hemorrhagic shock or postoperative urinary leakage, they were only seen in our malignant group with standard-RAPN. Though there was no statistical difference in complications just like in the previous studies, it may be explained by the already low rate of complications in RAPN [12,32]. Furthermore, other cases with complicated anatomic renal structures, such as horseshoe, solitary, or pelvic kidneys, or duplex kidneys with non-functioning symptomatic upper or lower moiety in pediatric patients, can also warrant ICG administration [12,21].

Second, ICG-RAPN can render renal tumors hypofluorescent (i.e., the tumor tissue is darker than normal) in near-infrared light, which allows for easy identification of the margin of the tumor and enables precision resection in real-time [8,9,16,20,21]. This method complements perioperative ultrasound, which is often inadequate because it cannot provide a real-time image due to the probe interfering when resection is performed [11]. The mechanism of hypofluorescence is based on ICG binding to a transmembrane protein called bilitranslocase (BLT), which expresses high concentrations in normal renal proximal and distal tubules (but not in glomeruli) and emits fluorescent light [33,34]. Previous studies have suggested that cells of AML and RCC do not express BLT and do not store ICG intracellularly [8,16]. Thus, their hypofluorescence is visible under NIRF, simplifying the differentiation of cancer cells from normal renal parenchyma. The aforementioned advantages have allowed ICG-RAPN to significantly shorten the WIT in most studies (WIT range: 11.5–24 min), and our study’s malignant group also displayed this trend finding despite the difference with the standard RAPN group being nonsignificant (24 vs. 26 min) [12]. This was critical because a WIT longer than 25 min is an independent predictor of newly diagnosed chronic kidney disease (CKD) in patients with a solitary kidney [26]. Finally, ICG-RAPN allows the blood supply of the remaining renal parenchyma to be evaluated after renorrhaphy because renorrhaphy itself can result in ischemic injury, especially in patients who already have CKD [11,15,35,36].

Nevertheless, ICG-RAPN has some limitations, such as being more suitable for superficially localized tumors due to its limited tissue penetration [16,23]. To increase its penetration depth and allow it to be performed in endophytic renal masses, the ICG was mixed with lipiodol, which was used to avoid a quick ICG washout from the renal tumor, in a previous study. Preoperative superselective transarterial delivery of the lipiodol–ICG mixture was initially employed, and then a postprocedural CT scan was completed for localization [11]. This method of using RAPN to deal with a completely endophytic tumor demonstrated acceptable renal functional outcomes and a lack of intraoperative and postoperative complications [11]. Another limitation of ICG-RAPN is that it offers the surgeon only one opportunity to evaluate the renal blood supply because once ICG is applied, it remains in the circulation system [15]. Furthermore, although none of our patients presented adverse effects from ICG usage, a previous study reported some rare allergic reactions associated with ICG usage in a population without known allergy-related vulnerabilities [37].

Our results suggested that ICG-RAPN’s advantage over standard RAPN in functional outcomes seemed to decrease with time because the postoperative eGFR preservation at 6 months was lower than at 3 months in both the benign and malignant groups. A similar phenomenon was also reported in a previous study [19]. Two possible mechanisms may explain the significant long-term decrease in renal function observed in our study. First, the greater recovery from kidney injury in patients who received standard RAPN (because they are more susceptible to acute tubular necrosis owing to increased renal ischemia) compared with the ICG-RAPN group at 6-month follow-up may explain their greater decrease from day 1 postoperative eGFR. Second, the compensation of the normal contralateral kidney (a feature in most patients of both groups and none of our patients had solitary kidney) for loss of renal function may have partially normalized the eGFR over the time leading up to the 6-month follow-up. Although superior short-term preservation of eGFR in the ICG-RAPN group may have implied less renal parenchyma ischemia, this benefit could have been masked over time by the compensation of the normal contralateral kidney, minimizing the difference between ICG and standard groups [38]. Therefore, for patients with a normal contralateral kidney, serum eGFR may not be a sensitive measure of ICG-RAPN’s value. Some studies have suggested the use of renal scan, CT scan with 3D volume rendering, or magnetic resonance renography to estimate the single residual renal volume more accurately [14,15,19].

Our study entailed several limitations. First, the relatively small number of patients and the retrospective analysis may have made the study susceptible to underrepresentation bias. In both the benign and malignant tumor groups, ICG-RAPN exhibited superior short- and long-term renal functional outcomes relative to standard RAPN. Although comparing the benign and malignant tumor groups one by one suggested only a nonsignificant advantage of ICG-RAPN over standard RAPN, a significant advantage in short-term renal functional outcomes was noted when we combined the benign and malignant tumor groups together. Therefore, if the number of patients is increased, the intraoperative and postoperative outcomes may reach significance. Second, because the individual surgeons determined whether to perform ICG-RAPN, selection would have been strongly biased toward standard RAPN. Third, serum eGFR could not reveal the exact residual kidney function and the contralateral normal kidney may have hampered the estimation of residual kidney function. Incomplete laboratory data and image examinations during follow-up might also have occurred, and different routines of the various surgeons may have resulted in a failure to accurately reflect all the outcomes.

## 5. Conclusions

In summary, although patients who underwent ICG-RAPN had longer operative times, they demonstrated superior short-term renal functional outcomes relative to those who received standard RAPN. For patients with malignant renal tumors, ICG-RAPN resulted in less operative blood loss than standard RAPN without increased positive surgical margin rates. Therefore, ICG-RAPN is an ostensibly safe procedure with potentially superior short-term renal functional outcomes. Further prospective randomized controlled trials are required to confirm whether this technique effectively provides the discussed intraoperative and functional advantages.

## Figures and Tables

**Table 1 cancers-14-03032-t001:** Baseline and clinical data of participants.

Variable	Total (*n* = 127)		
Patients, n	ICG (21)	No ICG (106)	*p*-Value
Age (years), median (IQR)	58 (42–67)	57 (49–66)	0.343
Male, n (%)	12 (57)	60 (57)	0.964
BMI (kg/m^2^), median (IQR)	26.5 (23.2–29.2)	25.1 (23.1–28.1)	0.219
Hypertension, n (%)	8 (38)	51 (48)	0.4
Diabetes, n (%)	4 (19)	30 (28)	0.59
ASA score, median (IQR)	2 (2–2)	2 (2–2)	0.819
ASA score, n (%)			0.933
1	4 (19)	25 (24)	
2	15 (71)	69 (65)	
3	2 (10)	12 (11)	
Left side, n (%)	8 (38)	43 (41)	0.833
RENAL score, median (IQR)	8 (6–8)	8 (6–9)	0.763
Tumor complexity, n (%)			0.78
Low (4–6)	7 (33)	28 (26.9)	
Intermediate (7–9)	12 (57)	63 (60.6)	
High (10–12)	2 (10)	13 (12.5)	
Tumor diameter in CT/MRI (cm), median (IQR)	3.5 (2.7–7.3)	3.2 (2.5–4.8)	0.239
Preoperative Hb (g/dL), median (IQR)	14.4 (12.7–16.0)	13.7 (12.5–15.0)	0.188
Preoperative eGFR (mL/min/1.73 m^2^), median (IQR)	91.9 (74.3–109.1)	91.5 (74.4–110.5)	0.731

Abbreviations: BMI = body mass index; CT = computed tomography; eGFR = estimated glomerular filtration rate; Hb = hemoglobin; ICG = indocyanine green; IQR = interquartile range; *n* = number; MRI = magnetic resonance imaging; RAPN = robot-assisted partial nephrectomy.

**Table 2 cancers-14-03032-t002:** Intraoperative and postoperative data of participants.

Variable	Total (*n* = 127)		
Patients, n	ICG (21)	No ICG (106)	*p*-Value
Operative time (min), median (IQR)	311 (263–360)	271 (217–310)	0.006
Estimated blood loss (mL), median (IQR)	50 (30–200)	100 (50–200)	0.09
Warm ischemia time (min), median (IQR)	21 (16–27)	24 (17–35)	0.25
Tumor size (cm), median (IQR)	3.3 (2.5–5.8)	2.9 (2.3–4.1)	0.174
Stay length (day), median (IQR)	7 (6–8)	7 (6–8)	0.545
Positive surgical margins, n (%)	2 (11)	8 (8)	0.66
Postoperative complications, n (%)	4 (19)	25 (24)	0.781
Minor (Clavien-Dindo I–II)	3 (14)	20 (19)	0.902
Major (Clavien-Dindo III–IV)	1 (5)	5 (5)	
Clavien-Dindo III ≥ 3, n (%)	1 (5)	5 (5)	1
One-year recurrence, n (%)	0 (0)	3 (4)	1
Trifecta achievement, n (%)	9 (56)	44 (46)	0.462
Post-op Hb at day one (g/dL), median (IQR)	12.9 (11.1–14.4)	12.2 (10.8–13.6)	0.223
Preservation rate of post-op Hb at day one (%), median (IQR)	89 (84–92)	89 (86–95)	0.398
Post-op eGFR at day one (mL/min/1.73 m^3^), median (IQR)	81.0 (64.0–93.8)	68.6 (51.5–93.9)	0.296
Preservation rate of post-op eGFR at day one (%), median (IQR)	84 (70–96)	79 (68–90)	0.2
Post-op eGFR at 3-months (mL/min/1.73 m^3^), median (IQR)	74.3 (62.8–91.7)	77.7 (65.1–94.0)	0.762
Preservation rate of post-op eGFR at 3-months (%), median (IQR)	90 (85–97)	85 (77–91)	0.031
Post-op eGFR at 6-months (mL/min/1.73 m^3^), median (IQR)	71.5 (65.6–94.6)	76.1 (64.9–90.1)	0.735
Preservation rate of post-op eGFR at 6-months (%), median (IQR)	83 (80–91)	81 (74–94)	0.346

Abbreviations: eGFR = estimated glomerular filtration rate; Hb = hemoglobin; ICG = indocyanine green; IQR = interquartile range; *n* = number; RAPN = robot-assisted partial nephrectomy; RCC = renal cell carcinoma.

**Table 3 cancers-14-03032-t003:** Intraoperative and postoperative data of benign and malignant tumors group.

Variable	Benign (*n* = 38)			Malignant (*n* = 89)		Variable
Patients, n	ICG (10)	No ICG (28)	*p*	ICG (11)	No ICG (78)	*p* Value
Operative time (min), median (IQR)	325 (275–396)	228 (187–291)	0.001	298 (247–350)	280 (222–325)	0.25
Estimated blood loss (mL), median (IQR)	200 (50–350)	90 (50–200)	0.272	30 (20–50)	100 (50–200)	<0.001
Warm ischemia time (min), median (IQR)	20 (16–25)	20 (15–29)	0.722	24 (15–28)	26 (18–36)	0.207
Tumor size (cm), median (IQR)	5.6 (2.3–9.2)	3.8 (2.7–5.5)	0.36	2.9 (2.5–3.7)	2.7 (2.2–3.7)	0.447
Stay length (day), median (IQR)	8 (6–8)	7 (6–8)	0.317	6 (5–8)	7 (6–8)	0.074
Positive surgical margins, n (%)	1 (13)	4 (18)	1	1 (9)	4 (5)	0.491
Post-op complications, n (%)	2 (20)	4 (14)	0.644	2 (18)	21 (27)	0.721
Minor (Clavien-Dindo I–II)	2 (20)	4 (14)	0.644	1 (9)	16 (21)	0.627
Major (Clavien-Dindo III–IV)	0 (0)	0 (0)		1 (9)	5 (6)	
Clavien-Dindo III ≥ 3, n (%)	0 (0)	0 (0)		1 (9)	5 (6)	0.558
One-year recurrence, n (%)	0 (0)	2 (8)	1	0 (0)	1 (2)	1
Trifecta achievement, n (%)	6 (67)	10 (48)	0.44	3 (43)	34 (46)	1
Post-op Hb at day one (g/dL), median (IQR)	11.5 (10.9–13.3)	11.5 (11.0–12.2)	0.66	13.7 (11.4–15.2)	12.5 (10.8–14.0)	0.1
Preservation rate of post-op Hb at day one (%), median (IQR)	87 (83–90)	89 (86–93)	0.351	91 (84–96)	90 (86–95)	0.985
Post-op eGFR at day one (mL/min/1.73 m^3^), median (IQR)	80.0 (66.0–119.1)	89.2 (71.7–114.1)	0.66	81.0 (39.9–85.1)	64.2 (45.0–84.4)	0.48
Preservation rate of post-op eGFR at day one (%), median (IQR)	93 (75–98)	85 (79–99)	0.961	77 (68–97)	76 (65–87)	0.313
Post-op eGFR at 3-months (mL/min/1.73 m^3^), median (IQR)	73.5 (65.1–139.6)	98.6 (83.7–121.3)	0.775	75.0 (56.3–84.4)	73.9 (61.2–90.1)	0.736
Preservation rate of post-op eGFR at 3-months (%), median (IQR)	93 (86–111)	86 (80–89)	0.095	88 (81–94)	85 (76–92)	0.201
Post-op eGFR at 6-months (mL/min/1.73 m^3^), median (IQR)	97.1 (69.2–98.8)	85.9 (67.3–100.0)	0.787	68.5 (54.3–83.3)	73.8 (64.2–88.1)	0.348
Preservation rate of post-op eGFR at 6-months (%), median (IQR)	84 (79–91)	80 (72–96)	0.516	83 (80–95)	81 (74–94)	0.448

Abbreviations: eGFR = estimated glomerular filtration rate; Hb = hemoglobin; ICG = indocyanine green; IQR = interquartile range; *n* = number; post-op = postoperative.

**Table 4 cancers-14-03032-t004:** High-grade complications and subsequent interventions.

Complication (Clavien-Dindo ≥ 3)	Benign		Malignant		Intervention
Patients, *n*	ICG (0)	No ICG (0)	ICG (1)	No ICG (5)	ICG (0)
Renal pseudoaneurysm, *n* (%)	0	0	1 (100)	3 (60)	Transarterial embolization
Hemogenic shock, *n* (%)	0	0	0	1 (20)	Emergent re-open surgery
Urine leakage with urinoma and UPJ obstruction, *n* (%)	0	0	0	1 (20)	Stent placement

## Data Availability

All data can be found in the text.

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
