# Peer review of "Clinical Benefits of Indocyanine Green Fluorescence in Robot-Assisted Partial Nephrectomy"

_cancers, 2022, doi:10.3390/cancers14123032_

Round 1
Reviewer 1 Report
Indocyanine green (ICG) is a substance that emits fluorescence when illuminated with light in the near-infrared spectrum (750-810 nm); bound to proteins can be visualized through tissues up to 1 cm thick. Used for various methods of imaging and diagnostic it is also used in the urologic field, during surgery.
The method consists of the intravenous injection of such contrast. In the operating room, using dedicated light sources, contrast allows better identification of structures by guiding the surgeon in dissection, with applicability in open surgery, video laparoscopy and robotic surgery.
Comment to Authors
Authors should be congratulated for the interesting topic discussed.
This paper aims to compare the intraoperative and postoperative outcomes and the differences in the results of ICG administration between patients with benign and malignant tumors.
Authors performed a retrospective study of clinical data of 127 patients from 2017 to 2020 who underwent robot-assisted partial nephrectomy (RAPN). Among them, 21 received ICG administration. 38 patients received a diagnosis of benign tumors, while 89 were diagnosed with malignant neoplasm.
The manuscript is well-written and easily readable, tables and graphics could be ameliorated allowing an easier synoptic view, but it is lacking in several points that would add value to the entire manuscript:
I suggest to consider the following manuscript (https://doi.org/10.3389/fped.2020.00314, https://doi.org/10.1016/j.jpurol.2020.07.008) the report preliminary experiences using ICG technology and include an important number of procedures that underline results achieved in this paper and ensure a significant increase in its scientific resonance.
Author Response
Dear Reviewer,
On behalf of the authors of this manuscript, we thank you very much for your thorough review of this manuscript. Your suggestions were very thoughtful and instructive. You identified some important concerns that we have never considered while making this study. We revised the manuscript according to the comments of yours point-by-point as follows. We hope that the revised manuscript will retain your attention and you will judge the revised manuscript to be suitable for publication in " Cancers".
The manuscript has been edited with the ‘’highlight’’ feature in Microsoft Word so that revisions may be clearly identified. Please forward the revised manuscript and reconsider the work for publication.
Yours sincerely,
First author: Yu-Kuan Yang
Corresponding author: Kai-Jie Yu and See-Tong Pang
Point 1: Authors should be congratulated for the interesting topic discussed.
This paper aims to compare the intraoperative and postoperative outcomes and the differences in the results of ICG administration between patients with benign and malignant tumors.
Authors performed a retrospective study of clinical data of 127 patients from 2017 to 2020 who underwent robot-assisted partial nephrectomy (RAPN). Among them, 21 received ICG administration. 38 patients received a diagnosis of benign tumors, while 89 were diagnosed with malignant neoplasm.
The manuscript is well-written and easily readable, tables and graphics could be ameliorated allowing an easier synoptic view, but it is lacking in several points that would add value to the entire manuscript:
I suggest to consider the following manuscript (https://doi.org/10.3389/fped.2020.00314, https://doi.org/10.1016/j.jpurol.2020.07.008) the report preliminary experiences using ICG technology and include an important number of procedures that underline results achieved in this paper and ensure a significant increase in its scientific resonance.
Response 1:
We have added several points of view from these two studies to our manuscript as the suggestions. Thank you for your constructive comments.

Reviewer 2 Report
Indocyanine green seems to be an interesting diagnostic agent in differentiating healthy from pathological tissues during kidney surgeries. This problem was previously signalized by Mitsui Y, et al. (Indocyanine green (ICG)-based fluorescence navigation system for discrimination of kidney cancer from normal parenchyma: application during partial nephrectomy. Int Urol Nephrol. 2012. PMID: 22215306).
Remarks:
1) Authors through the whole manuscript mention that groups differed, but significance was not achieved; please clearly state what was significant in your study and especially analyze/discuss these results; small amount of patients in groups may affect results; please do not say about 'the trend' (line 248);
2) I would like to know more as a Reader about benefit regarding lower risk of surgical complications (Figure 4), please try to highlight that;
3) what about preservation of GFR at 3-months not later? did patients have reoccurrence of primary disease? or received (nephro)toxic treatment? any anticancer drugs?
4) how your study showed that RCC is associated with obesity, hypertension and CKD? (lines 207-208)
Author Response
Dear Reviewer,
On behalf of the authors of this manuscript, we thank you very much for your thorough review of this manuscript. Your suggestions were very thoughtful and instructive. You identified some serious and important concerns that we have never considered while making this study. We revised the manuscript according to the comments of yours point-by-point as follows. We hope that the revised manuscript will retain your attention and you will judge the revised manuscript to be suitable for publication in " Cancers".
The manuscript has been edited with the ‘’highlight’’ feature in Microsoft Word so that revisions may be clearly identified. Please forward the revised manuscript and reconsider the work for publication.
Yours sincerely,
First author: Yu-Kuan Yang
Corresponding author: Kai-Jie Yu and See-Tong Pang
Point 1: Authors through the whole manuscript mention that groups differed, but significance was not achieved; please clearly state what was significant in your study and especially analyze/discuss these results; small amount of patients in groups may affect results; please do not say about 'the trend' (line 248);
Response 1: A longer operative time (311 vs. 271 minutes; p = 0.006) but superior preservation of estimated glomerular filtration rate (eGFR) at 3-month follow-up (90% vs. 85%; p = 0.031) were observed in the ICG-RAPN group. Among the patients with malignant tumors, less estimated blood loss (30 vs. 100 mL; p < 0.001) was reported in the ICG-RAPN subgroup. We apologize for the misleading and confusing original statement. We have made a better description to avoid misunderstanding in our manuscript. Thank you for your comments. (lines 254, page 11)
Point 2: I would like to know more as a Reader about benefit regarding lower risk of surgical complications (Figure 4), please try to highlight that;
Response 2: For patients who had high-grade complications (Clavien–Dindo classification of ≥3), 1 of them underwent ICG-RAPN and 5 of them underwent standard-RAPN. The benefit of using ICG-RAPN is to identify the arterial blood supply of renal tumors more clearly (especially in malignant tumors since they tend to have more accessory and complex blood supplies that are difficult to be thoroughly identified with preoperative imaging) and resect the renal tumors more precisely in real time since tumors are cold areas under near-infrared light, allowing for easy identification of the margin. By the visual advantages, we could avoid injuries to blood vessels and minimize the sacrificed renal parenchyma during tumor resection, and therefore lower the risk of high-grade surgical complications, for example, intra-operative massive bleeding, renal pedicle injury resulting in total resection of the kidney and postoperative renal pseudoaneurysm or urinary leakage. In other words, ICG could be the simplest way to achieve the goal of nephron-sparing robotic surgery efficiently. Thank you for your comments.
Point 3: what about preservation of GFR at 3-months not later? did patients have reoccurrence of primary disease? or received (nephro)toxic treatment? any anticancer drugs?
Response 3: Among the patients with malignant tumors, none of them had reoccurrence of primary disease within one year and none of them had received (nephro)toxic treatment or any anticancer drugs in the ICG-RAPN subgroup. For postoperative eGFR preservation at 6 months was lower than at 3 months in both the benign and malignant groups, a similar phenomenon was also reported in a previous study (McClintock et al. 2014). Two possible mechanisms may explain the significant long-term decrease in renal function observed in our study. First, the greater recovery from kidney injury in patients who received standard RAPN compared with the ICG-RAPN group at 6-month follow-up may explain their greater decrease from day 1 postoperative eGFR. Second, the compensation of the normal contralateral kidney (a feature in most patients of both groups and none of our patients had solitary kidney) for loss of renal function may have partially normalized the eGFR over the time leading up to the 6-month follow-up, minimizing the difference between ICG and standard groups. We have also discussed this in our manuscript. Thank you for your constructive comments. (lines 275-290, page 12)
Point 4: how your study showed that RCC is associated with obesity, hypertension and CKD? (lines 207-208)
Response 4: In this cohort, we made a comparison in demographics between patients underwent surgery with or without ICG assistance and also a comparison with peri-operative variables between malignant and benign renal tumor group. However, we didn’t analyze the difference between malignancy and normal cohort. We found that more male patients (67 vs. 5; p < 0.001), more patients with hypertension (49 vs. 10; p = 0.003) and lower preoperative eGFR (88.9 vs. 104.8; p = 0.011) were seen in the malignant group. Although in the comparison between benign and malignant tumors, no significant differences in preoperative BMI (24.9 vs. 25.0; p = 0.072) was noted, we didn’t include normal patients as control. We couldn’t make a conclusion about the BMI difference between malignancy and normal groups. According to our previous study (DOI: 10.2147/TCRM.S164592), we indeed explored the demographics of RCC patient cohort which was consistent to the reference we had. We have revised the description for a better understanding. Thank you for your comments. (lines 206-208, page 10)

Round 2
Reviewer 1 Report
Authors answered all comments and suggestions.
Reviewer 2 Report
Authors answered to my questions, however if they added more corrections (as written in responce) should highlight them in the text.